# Thermal Sensitivity of a Microoptoelectromechanical Evanescent-Coupling-Based Accelerometer

**DOI:** 10.3390/s25206388

**Published:** 2025-10-16

**Authors:** Evgenii Barbin, Ivan Kulinich, Tamara Nesterenko, Alexei Koleda, Ayan Myrzakhmetov, Denis Mokhovikov, Sergey Vtorushin, Alena Talovskaia

**Affiliations:** 1Research Institute of Microelectronic Systems, The Tomsk State University of Control Systems and Radioelectronics, 634050 Tomsk, Russiakoledaan@tpu.ru (A.K.); alena.a.talovskaia@tusur.ru (A.T.); 2Laboratory of Radiophotonics, V.E. Zuev Institute of Atmospheric Optics of SB RAS, 634055 Tomsk, Russia; 3Division for Electronic Engineering, National Research Tomsk Polytechnic University, 634050 Tomsk, Russia

**Keywords:** accelerometer, optical transducer, waveguide, directional coupler, coupling length, temperature sensitivity, transmission coefficient, coefficient of thermal expansion, refraction index

## Abstract

This treatise studies the thermal sensitivity of the mechanical and optical transmission coefficients of a microoptoelectromechanical (MOEM) accelerometer based on evanescent coupling in a temperature range from minus 40 to plus 125 °C. Two types of optical measuring transducers are considered: based on a directional coupler and a resonator. This analysis covers the optical and mechanical components of the thermal sensitivity of the transmission coefficient. In terms of the mechanical part, the temperature effect induces changes to the linear dimensions of the structure and material characteristics and causes internal mechanical stresses as well. The temperature effect on the optical system of the accelerometer is conditioned by the thermo-optic effect of the materials the optical waveguides are made of. This study includes experiments on the refraction index dependence on the temperature of the optical films that compose the optical system of the MOEM accelerometer. The experiment shows that the refraction index of the films grows with temperature and amounts to 0.12642 ppm/°C for silicon nitride on the SiO_2_/Si substrate. For the optical measuring transducer based on a directional coupler, the thermal sensitivity of the accelerometer’s optical transmission coefficient is 580 ppm/°C. For the resonator-based transducer, the thermal sensitivity is 0.33 °C^−1^. The thermal sensitivity of the normalized mechanical transmission coefficient of the accelerometer is 120 ppm/°C. For optical measuring transducers based on a directional coupler, the contribution of the temperature dependent refraction index alteration to the overall error is 5 times larger than that of the MOEM accelerometer’s mechanical parameters, while for the resonator-based transducer the difference reaches 3000 times. This means its operability is only possible in a thermostatic environment.

## 1. Introduction

Nowadays, MEMS accelerometers are widely used to measure acceleration in different industries: vehicle monitoring, seismology, aerospace structures, large structure monitoring (bridges, masts, drilling rigs, power lines, etc.) and navigation systems [1,2,3,4,5].

Almost any accelerometer contains a proof mass (PF) mounted on a spring suspension. Under acceleration, the PF displaces in relation to the housing. The mechanical displacement of the PF proportional to the acceleration can be measured by different methods: capacitive, piezoelectric, optical [6,7,8,9,10,11,12,13,14]. Among measuring transducers, the capacitive method is the most widely implemented due to its simplicity, compatibility with integrated circuits and sufficient performance in the majority of fields. However, this method has a number of drawbacks, such as susceptibility to electromagnetic noise, measurement errors caused by the edge effect and stray capacitance, which prohibit the application of such accelerometers in the measurement of micro accelerations.

Another type of MEMS accelerometer is the microoptoelectromechanical (MOEM) accelerometer in which the displacement of the PF changes the parameters of the light flux. The optical method enables measurement of the PF displacements corresponding to very small accelerations. There are a number of techniques for such measurements: registration of light flux intensity alteration, fiber Bragg grating (FBG), Fabry–Pérot interferometry, Talbot effect [15,16,17,18,19,20], etc. MOEM accelerometers possess such strong points as high accuracy and operability under strong electromagnetic interference. Therefore, they are a promising solution for inertial navigation, geophysics and other fields that demand high sensitivity and accuracy; they are also a basis for the development of advanced accelerometers.

However, the accelerometers are frequently required to work in wide temperature ranges: from minus 40 up to plus 80, 105 or even 175 °C. Such temperature swings affect the mechanical, optical and electrical parts of the accelerometer, which requires enhanced materials and design. This should be taken into account when developing MOEM devices.

In general, there are two main methods of temperature compensation: hardware and software compensation. The hardware approach assumes the selection of materials and design and implementation of thermostating via Peltier elements or heaters. The latter is more resource consuming due to increased volume, mass and energy consumption of the final product. Hence, the hardware method is inapplicable in the cases demanding low cost and energy consumption. In the second method—software method—the relation between the temperature and output signal of the accelerometer is studied in the first place; then, a model of temperature drift is established for its compensation.

The temperature effect in a MOEM accelerometer can be divided into separate effects. First, there is the alteration of the mechanical part of the accelerometer (Young’s modulus change, thermal deformation, thermally induced stresses). Second, there is the alteration of the characteristics of the accelerometer’s optical part.

The influence of temperature on the mechanical part of the transducer is similar for all MEMS accelerometers implementing different methods of PF displacement measurement. For instance, in a resonance MEMS accelerometer, the output signal temperature drift can be eliminated via the design of the transducer crystal consisting of two identical tuning forks with opposite sensitivity axes [21]. The transducer continuously tracks the resonance frequency on temperature change; both tuning forks possess the same temperature sensitivity and opposite sensitivity to external acceleration. Differential signal processing identifies the difference in the frequencies of two resonance accelerometers, which enables acceleration measurement with the temperature drift being compensated. In work [22], temperature testing was conducted to obtain initial data and derive a low-order polynomial equation. The equation was combined with the measurement error associated with thermal drift, after which they were used to compensate the accelerometer’s temperature drift. After the compensation, the mean square errors calculated for axes *X* and *Y* have decreased by 96%. In [23], Shen et al. proposed new multi-input/single-output (MISO) model based on a genetic algorithm (GA) and Elman neural network to increase the modeling accuracy for the transducer’s temperature drift. Work [24] considered an algorithm for compensating the accelerometer’s temperature error that was based on a neural network which compensated all measurements at once without modeling the error parameters (displacement, scale factor, axial misalignment) separately. The method of polynomial regression allowed decreasing the accelerometer’s error by 69%, while the neural network reached 99%.

Paper [25] proposed a method for temperature compensation of output signal errors of a MOEM accelerometer based on grating interferometric cavity. At different temperatures, a deformation occurred that changed the distance between the grating and the transducer chip, which in turn caused the modulation of optical signal intensity. Temperature compensation using a static temperature model was adjusted by modeling when the temperature loads changed from minus 20 to plus 80 °C. The standard deviation of the output intensities of the MOEMS accelerometer at different temperatures after compensation amounted to 0.0036.

Work [26] studied the effect of thermal load that occurs during packaging on the MOEM accelerometer’s characteristics and that negatively affects the dynamic characteristics and life of the accelerometer. The results showed that adequate correspondence of the thermal expansion coefficient (TEC) of the substrate and transducer structure, reduced elasticity modulus and increased thickness of the adhesive layer may considerably drop the thermal stress. Moreover, well-designed housing can reduce the zero drift and scale factor drift to minimum. Small TEC of the substrate also effectively reduces the temperature effect.

The authors of [27] identified a heat source in MOEM accelerometers induced by light transmission along the waveguide mounted on the PF. As a result, the heat flux from the PF propagated to the substrate via spring suspensions. Therefore, the temperature profile of the structure changed under laser radiation together with thermal convection due to the environment temperature. Finally, this caused deformation, stresses and non-stationary state. These temperature effects were also affected by different geometry and dimensions (length, width, thickness) of the suspensions. On the other side, the design and dimensions of the elastic elements of the suspension determined the mechanical characteristics of the accelerometer (sensitivity and resonance frequency). The work unveiled the correlations between the mechanical and thermal characteristics, which allowed developing the recommendations for a MOEM accelerometer design with decreased thermally induced deformation and rise time without degradation of other mechanical characteristics.

Work [28] described a two-axis MOEM accelerometer with Bragg gratings. The applied differential measurement between the fiber Bragg gratings in a multicore fiber enabled thermally insensitive acceleration measurements.

Paper [29] proposed a method for designing a thermally stable MOEM accelerometer based on optical interference. To minimize the thermal drift, the authors suggested a multi-layer structure. The correlation between the material properties and thermal deformation were obtained. The structure comprised three layers, namely substrate, sensitive structure and cap, whose materials were silicon, silicon and glass, respectively. In the substrate and cap, there was a groove to ensure free movement of the proof mass. In the substrate’s groove, a temperature control system was located that was made of platinum and contained a heating resistor.

In work [30], a thermally insensitive MOEMS accelerometer based on the waveguide Bragg grating was proposed. The transducer consisted of two waveguide gratings mounted on a spring suspension of the proof mass. The gratings were aligned on beams in a way so that they had different linear expansion coefficients depending on their location on a beam. As a result, the external acceleration under changing temperature could be accurately measured by the shift of the Bragg wavelength for both the gratings.

Following the analysis of papers presented in the overview, one can conclude the existence of studies on the temperature change of the characteristics of the mechanical and optical parts of an accelerometer on the basis of Bragg gratings and optical interference. However, we could not find works studying the temperature properties of MOEM accelerometers based on directional couplers and ring resonators, which confirms the relevance and novelty of our research.

The current paper studies the influence of temperature on the dynamic characteristics of an evanescent coupling-based MOEM accelerometer. The goal of this study is to determine the thermal sensitivity of the mechanical and optical transmission coefficients of the MOEM accelerometer and to compare two variants of optical measuring transducers (OMTs): based on a directional coupler and a resonator.

## 2. Experiments and Modeling of the Temperature Influence on the MOEM Accelerometer

### 2.1. Functional Scheme of the Accelerometer

In MOEM accelerometers, the displacement of the proof mass is measured by different optical methods [31]. One of the most promising optical methods for measuring micro accelerations that is highly integrated on the chip level is the implementation of evanescent coupling that is used in elements of photonic integrated circuits (PICs).

The working principle of such MOEM accelerometer is based on the measurement of the optical flux induced by the proof mass displacement under measured acceleration. The accelerometer contains micromechanical, optical and electronic parts (Figure 1) [32].

Figure 2 depicts two functional schemes of MOEM accelerometers that contain movable waveguide 1, represented by a directional coupler (Figure 2a) or a ring resonator (Figure 2b), that are mounted on proof mass 3. The proof mass is mounted on spring suspension 4 in housing 5. Fixed waveguide 2 is coupled with housing 5. Between the movable and fixed waveguides there is an air gap (200–500 nm) sufficient for the optical power to begin flowing—due to the evanescent-wave coupling effect—from the fixed waveguide into the movable one where it dissipates. The optical modes and the coupling effect at different gaps will be shown below in Section 2.3.

Optical radiation from the laser diode is fed through the input port to the fixed waveguide and extracted through the output port to the photodiode. If acceleration *a_y_* is present, the proof mass displaces along *Y* axis (sensitivity axis). The gap between the movable and fixed waveguides changes, which alters optical power *P_opt_* that is proportional to measured acceleration *a_y_*. Therefore, accelerometer Figure 2a,b differ in the OMT type, while having identical mechanical parts.

Total transmission coefficient (TTC) of the accelerometer is calculated as(1)$$K_{acc}=K_{mech}\cdot K_{opt}$$
where *K_opt_* is the transmission coefficient of the OMT, and *K_mech_* is the transmission coefficient of the mechanical part of the accelerometer.

In this connection, we can investigate the temperature influence both on the mechanical design of the accelerometer and OMT separately and then compare the two OMTs.

### 2.2. Temperature Influence on the Mechanical Transmission Coefficient of the Accelerometer

The dynamic characteristics of the accelerometer’s mechanical system can be obtained from the following equation:(2)$$\ddot{y}+\frac{\omega_{y}}{Q_{y}}+\omega_{y}^{2}y=a_{y}$$
where *m*, *Q_y_*, $\omega_{y}=\sqrt{\frac{k_{y}}{m}}$ are the mass, *Q*-factor and resonance frequency of the micromechanical structure; *k_y_* is the stiffness of the spring suspension.

The main characteristics of the mechanical system of a MOEM accelerometer are as follows: dimensions, eigenfrequencies along sensitivity axis *Y* and cross-axis *Z* and the mechanical transmission coefficient that is calculated as(3)$$K_{M}=\frac{y}{a_{y}}=\frac{1}{\omega_{y}^{2}}$$

High resonance frequency necessary to increase the accelerometer bandwidth leads to very small displacements that should be measured by the optical transducer with high resolution. Balance should be found between the resolution and bandwidth of the accelerometer.

Temperature changes the linear dimensions of the structure, the elasticity modulus of silicon and Poisson’s ratio. The structure suffers the occurrence of internal mechanical stresses due to different thermal expansion coefficients of the coupled materials; the geometry deteriorates, and viscous friction forces change. All these factors change the frequency properties of the mechanical system; the *Q*-factor and total mechanical transmission coefficient of the accelerometer change in accordance with Equation (3).

The largest effect temperature exerts is on the elastic elements of the proof mass suspension [33]. The mechanical stresses arising due to different coefficients of linear temperature expansion (CLTE) of the coupled materials are the main reason for altering eigenfrequency and, consequently, the mechanical transmission coefficient of the accelerometer.

The accelerometer’s mechanical system (Figure 3) consists of a movable part (proof mass) and fixed part (wafer, anchors, contact pads) made of different layers of materials. There is an air gap between the proof mass and the wafer that ensures free movement of the proof mass.

To study the temperature influence, a finite element method (FEM) was implemented in ANSYS software (2022 R1, Ansys, Inc., Canonsburg, PA, USA). The temperature of the anchor binding with BCB was 250 °C, corresponding to the curing temperature, while the surrounding temperature was altered from minus 40 to plus 125 °C. The model took into account temperature-induced changes to the materials (Young’s modulus, Poisson’s ratio, CLTE). The displacement of the silicon wafer of the mechanical system along *Z* axis was prohibited, which enabled its free expansion along the *X* and *Y* axes. Maximum strain and stresses of the mechanical system were observed at minus 40 °C (Figure 4).

Maximum mechanical stresses *σ_S_* in the structure occurred at minus 40 °C and amounted to 123.65 MPa for the gold layer (Figure 5) used for consequent wiring with the housing. These stresses did not exceed the ultimate tensile strength of gold (140 MPa).

Maximum mechanical stresses *σ_S_*_BCB (Figure 6) of 60.797 MPa at minus 40 °C occurred in the BCB layer at the mounting spot of fixed anchors and did not exceed the material’s ultimate tensile strength of 87 MPa.

Maximum mechanical stresses *σ_S_* in the proof mass (Figure 6) occurred at minus 40 °C and amounted to 1.936 MPa. Therefore, the temperature effect on the MOEM accelerometer in the range from minus 40 to plus 125 °C could not cause failures in its mechanical part. However, the effect alters the linear dimensions of the sensor, elasticity moduli and Poisson’s ratios of the materials, which changes eigenfrequency *f*_1_ = *ω*/2*π* (Hz) (Equation (3)) of the accelerometer along sensitivity axis *Y* (Figure 7).

With eigenfrequency changes the mechanical transmission coefficient of the accelerometer (Equation (3)). At a 1-degree Celsius change in temperature, the temperature sensitivity changes as(4)$$SK_{mech}=\frac{K_{mech}^{\max}-K_{mech}^{\min}}{\left|TK_{mech}^{\max}\right|+\left|TK_{mech}^{\min}\right|}$$
where $K_{mech}^{\max}$, $K_{mech}^{\min}$ are the maximum and minimum values of the transmission coefficient of the mechanical part of the accelerometer at corresponding temperatures; $\left|TK_{mech}^{\max}\right|$, $\left|TK_{mech}^{\min}\right|$ are the maximum and minimum temperatures (°C).

The absolute temperature sensitivity of the mechanical transmission coefficient of the accelerometer per 1 °C amounted to 7.55 × 10^−11^ (Figure 8a).

The thermal sensitivity of the normalized transmission coefficient per 1 °C amounted to 0.00012 °C^−1^ or 0.012%/°C (Figure 8b).

### 2.3. Temperature Influence on the Optical Measuring Transducer

Temperature affects the optical characteristics of the materials, which changes the general and effective refraction index, phase shift in waveguides and resonance shift in ring resonators. All these factors cause errors [34,35,36,37]. Further, we consider two types of accelerometer’s OMTs: based on a directional coupler and a resonator [38].

#### 2.3.1. Experimental Determination of the Refraction Index of the Optical Waveguides

At the atomic level, the thermo-optic effect of the optical waveguide’s material is conditioned by several mechanisms. First, material heating changes its density, which affects the refraction index. Second, thermal expansion of the crystal lattice alters atomic spacing, which affects the electron configuration, hence the optical properties. Third, temperature change can alter electron distribution at different energy levels and sublevels in an atom or a molecule, which also affects optical absorption and refraction.

To obtain the numerical values of the temperature coefficients for refraction index change, experiments were performed on optical films (Si/SiO_2_/Si_3_N_4_) (Figure 9a) that form the optical part of the MOEM accelerometer using a spectroscopic ellipsometer EGS01-UI (Beijing Ellitop Scientific Co., Ltd., Beijing, China) (Figure 9b). To decrease the effect of mechanical stresses on the temperature plot, we performed quick annealing of the specimen in nitrogen at 600 °C for 180 s. We assumed that annealing decreases internal mechanical stresses in Si_3_N_4_/SiO_2_ films, while diffusion of hydrogen atoms in Si_3_N_4_ film to the surface and restoration of Si-N bonds make the films more optically stable [39].

Flat heating element 2 was placed on the ellipsometer. The plate was heated to 25–85 °C. The specimen temperature was altered by the heating plate, which was powered by power source 4 with voltage up to 24 V. Pyrometer 3 was used to remotely measure the temperature of specimen 5 in the ellipsometer’s measurement spot. Before each measurement, the height and levelness of the specimens in relation to the ellipsometer’s arms were adjusted using the object table. The measurements were made at wavelengths from 400 to 1600 nm and an angle between the arms of 140 degrees. Figure 10 and Figure 11 present the results of films’ refraction index measurement at a wavelength of 1550 nm.

The experimental temperature coefficient of the refraction index of the Si_3_N_4_ film on the SiO_2_/Si substrate amounted to 1.2642 × 10^−5^ °C^−1^.

The experimental temperature coefficient of the refraction index of the SiO_2_ film amounted to 0.14433 × 10^−5^ °C^−1^.

Following Figure 10 and Figure 11, the refraction index of the films increased with temperature, which corresponds to the literature data. The obtained results were used in further study of the accelerometer’s OMT. For silicon, the refraction index was assumed to be 0.95 × 10^−5^ °C^−1^.

#### 2.3.2. Measuring Transducer Based on a Directional Coupler

The OMT of an accelerometer [38,40] includes two basic optical elements: an optical waveguide and a directional coupler. In our study, the directional coupler (Figure 12) was represented by two coupled waveguides divided by an air gap. Denoted ports in Figure 12 are as follows: 1—Input port, 2—Through port, 3—Drop port and 4—Reflect port.

Optical transmission coefficient (OTC) for a particular design of a directional coupler depends on the difference between the effective refraction indexes of even and odd modes.(5)$$T_{opt}=\frac{P_{th}}{P_{in}}=\cos^{2}\left(\frac{\pi \cdot n_{effeven}\left(T\right)-n_{effodd}\left(T\right)}{\lambda }L\right)$$
where *λ* is the wavelength (nm), *T* is ambient temperature; *n_effeven_* (*T*) and *n_effodd_* (*T*) are temperature dependences of even and odd modes of the effective refraction indexes in the waveguides; *P_th_* and *P_in_* are, respectively, optical radiation power values on Through port 2 and Input port 1. Figure 13 depicts even and odd TE-modes of the coupled waveguides with an air gap of 360 nm.

The structure of the directional coupler was formed on the silicon substrate and consisted of 350 × 850 nm waveguides from silicon nitride surrounded by 2 μm layers of silicon oxide at coupling length L = 100 μm. Figure 14 shows the power maps of the input coupling structure with the gap ranging from 105 to 500 nm.

At a fixed coupling length and altering gap between the waveguides, the optical power manages to migrate to the drop port and back. This changes the optical transmission coefficient of the accelerometer (Figure 15).

The OTC of the measuring transducer in terms of displacement amounted to 6.779 × 10^6^ m^−1^ at an initial working gap of 360 nm and a dynamic range of ±80 nm.

To obtain the temperature dependence of the optical transmission coefficient of the OMT by the method of Finite-Difference Eigenmode (FDE), we studied the alteration of the waveguide’s effective refraction index *n_effwvg_* and effective refraction indexes of even (*n_effeven_*) and odd (*n_effodd_*) modes of the directional coupler with an air gap of 360 nm that corresponded to the half of the OTC. The temperature dependence of the refraction index was set by the numerical values of the refraction index obtained in experiments at a chosen temperature (linear approximation of data from Figure 11 for Si_3_N_4_ and from Figure 12 for SiO_2_) that were filled into the properties of corresponding materials. A series of calculations was performed in a temperature range from minus 40 to plus 100 °C with a step of 1 degree. For silicon, refraction index temperature sensitivity of 0.95 × 10^−5^ °C^−1^ was adopted. The results are presented in Figure 16.

The modeled temperature coefficient of effective refraction index alteration for single and coupled waveguides of the directional coupler amounted to 0.1214 × 10^−3^ °C^−1^.

Temperature change also alters the phase of optical radiation in the waveguide that is determined as(6)$$\varphi =\frac{2\pi L\left(n_{eff}-n_{eff20}\right)}{\lambda }$$
where *n_eff_* is the effective refraction index at temperature under study; *n_eff20_* is the effective refraction index at 20 °C. The alteration of the phase shift on temperature is depicted in Figure 17.

Since the temperature sensitivities of even and odd modes are different, the difference between the effective refraction indexes also changes, which alters the OMT’s optical transmission coefficient. The alteration of OTC on temperature for the Through and Drop ports at a gap of 360 nm (Figure 18) were calculated using Equation (5). *T_th_* in this case corresponds to *T_opt_* (5); *T_drop_* is given as reference.

Following the plots, in a temperature range from minus 40 to plus 100 °C, the OTC of the directional coupler-based accelerometer changed from 0.452043 to 0.534166. The thermal sensitivity of the OTC amounted to 0.00058 °C^−1^.

The sensitivity of the normalized mechanical transmission coefficient (MTC) per 1 °C amounted to 0.00012 °C^−1^. The dependence of the total transmission coefficient of the MOEMA on temperature is presented in Figure 17 and corresponds to expression(7)$$K_{OMTDC}=K_{mech}\cdot T_{opt}\left(1+K_{temp}\left(T-20\right)\right)$$
where *K_mech_* = 6.11 × 10^−7^ °C^2^ is the mechanical transmission coefficient; *T_opt_* = 6.779 × 10^6^ m^−1^ is the optical transmission coefficient (OTC); *K_temp_* = 7 × 10^−4^ °C^−1^ is total thermal sensitivity of the mechanical and optical transmission coefficients.

Following Figure 19, for the optical coupler, the contribution of refraction index changes on temperature is more than 5 times larger than that of the MOEMA’s mechanical parameters.

#### 2.3.3. Measuring Transducer Based on a Resonator

The second variant of the MOEMA’s measuring transducer is an optical ring resonator (Figure 20). The resonator can be represented as two symmetrically arranged directional couplers considered above. Gap G1 of the lower directional coupler can change under acceleration, while gap G2 of the upper directional coupler is fabricated with a required gap.

In resonators used as a PIC element, an important parameter is the free spectral range (FSR) that depends on the resonator’s perimeter. In the case of the resonator’s application as the OMT, this parameter is of no particular importance, so the resonator’s perimeter was minimized to 354 μm. The OTC of the resonator-based measuring transducer in terms of displacement amounted to 650 × 10^6^ m^−1^ at an initial working gap of 440 nm and a dynamic range of ±1.5 nm. Thus, the sensitivity of such transducer almost 100 times exceeds that of the directional coupler-based measuring transducer.

The influence of temperature on the OTC of the MOEMA’s resonator-based OMT was studied for an air gap of 360 nm. The analysis results demonstrated extremely high sensitivity of the resonator to temperature. Figure 21 depicts a limited temperature range of the resonator’s amplitude frequency response dependence on wavelength.

The resonance shift on temperature amounted to 0.14 nm/°C. Since the operation of the resonator-based measuring transducer requires a fixed wavelength, the alteration of the wavelength leads to a change of the OTC. Figure 22 presents the dependencies of the OTC on temperature for different temperature ranges at different wavelengths of OMT excitation.

Thus, resonance adjustment changes the transmission coefficient by 0.33 °C^−1^. The transmission coefficient changes from minimum to maximum in a range of (2–3) °C, while for the whole temperature range from minus 40 to 100 °C, it has several local minima. The temperature effect of mechanical transmission coefficient alteration (0.00012 °C^−1^) on the OMT’s total transmission coefficient, in this case, can be neglected.

## 3. Discussion

Following the study results, temperature appreciably affects the optical transmission coefficient of a MOEMA, especially a resonance one where thermal sensitivity of the optical transmission coefficient (0.33 °C^−1^) is nearly 3000 times higher than the sensitivity of the mechanical transmission coefficient (0.00012 °C^−1^). In this case, thermostatic environment and methods of output signal correction should be used, and temperature adjustment accuracy should be very high. For instance, to ensure temperature error of the TTC below 1%, the temperature stability of the thermostating system should lie within 0.03%, which is a nontrivial challenge.

To eliminate general temperature error of an accelerometer with a directional coupler-based OMT, the adjustment of the output signal using a temperature sensor is sufficient, because a general alteration of the transmission coefficient at the whole temperature range is below 10%.

It is worth mentioning that the investigation of temperature influence on the optical transmission coefficient did not account for the alteration of the refraction index on the internal stresses in the silicon nitride waveguides, because currently there are no proven methods for analytic estimation of real internal stresses, while these stresses can be reduced by varying the nitride film deposition parameters.

Another important factor affecting the alteration of the optical transmission coefficient on temperature is the variation of the relative positions of the movable and fixed waveguide due to the expansion of the wafer and proof mass materials. In an ideal design of the MOEMA, the calculations of deformation (Figure 4) have shown that the thermal sensitivity of the OTC in the directional coupler-based measuring transducer amounted to 0.0002 °C^−1^, while for the resonator-based transducer it was 0.02 °C^−1^. However, it should be taken into consideration that real values will differ because of the nonideality of materials, couplings, aligning and nonuniform stiffness of suspensions due to fabrication errors, etc., which requires additional research.

## Figures and Tables

**Figure 1 sensors-25-06388-f001:**
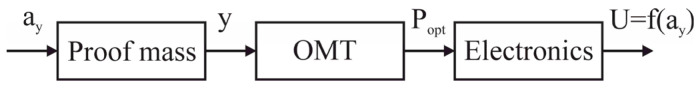
Structure of the MOEM accelerometer. Legend: a_y_ (m/s^2^)—measured acceleration; *y* (m)—proof mass displacement; *P_opt_* (W)—optical power at the output of the OMT; *U* (V)—output voltage.

**Figure 2 sensors-25-06388-f002:**
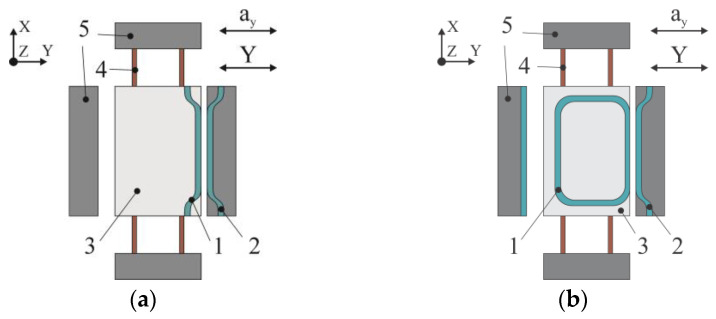
Functional scheme of the MOEM accelerometer: (**a**) directional coupler as the movable waveguide; (**b**) ring resonator as the movable waveguide.

**Figure 3 sensors-25-06388-f003:**
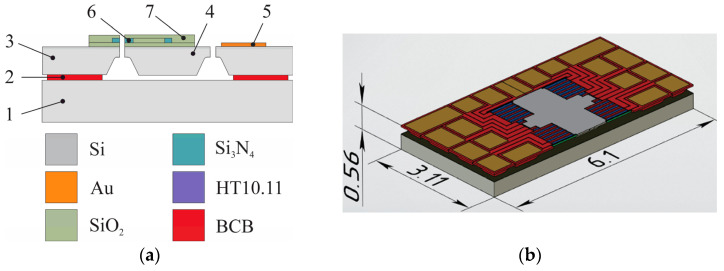
Accelerometer’s mechanical system: (**a**) structure of the accelerometer’s mechanical system; (**b**) 3D-model of the accelerometer. Legend: 1—350 μm silicon fixed wafer; 2—5 μm layer of CYCLOTENE 4000 (BCB) binding fixed wafer and anchors; 3—35 μm layer of silicon forming the anchors and the proof mass 4; 5—2 μm *gold* contact pads on anchors for connecting sensor wires in the housing; 6—silicon nitride waveguides surrounded by silicon oxide 7.

**Figure 4 sensors-25-06388-f004:**
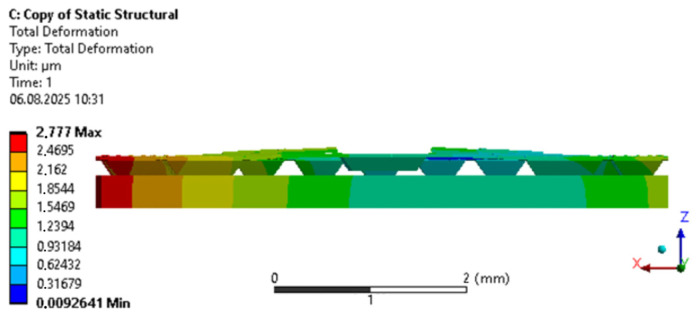
MOEMA deformation at minus 40 °C (front view).

**Figure 5 sensors-25-06388-f005:**
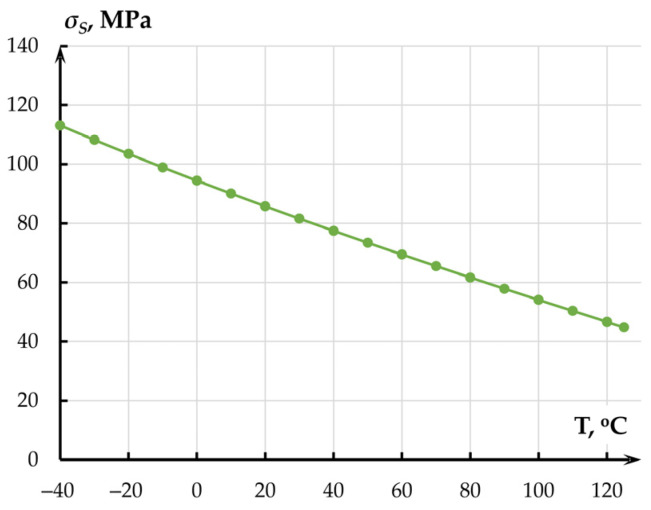
Dependence of mechanical stress on temperature in the gold layer.

**Figure 6 sensors-25-06388-f006:**
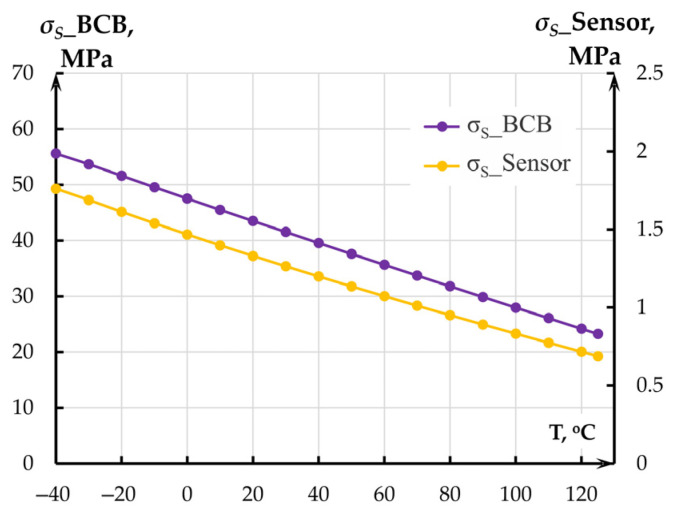
Mechanical stresses in silicon and BCB layer.

**Figure 7 sensors-25-06388-f007:**
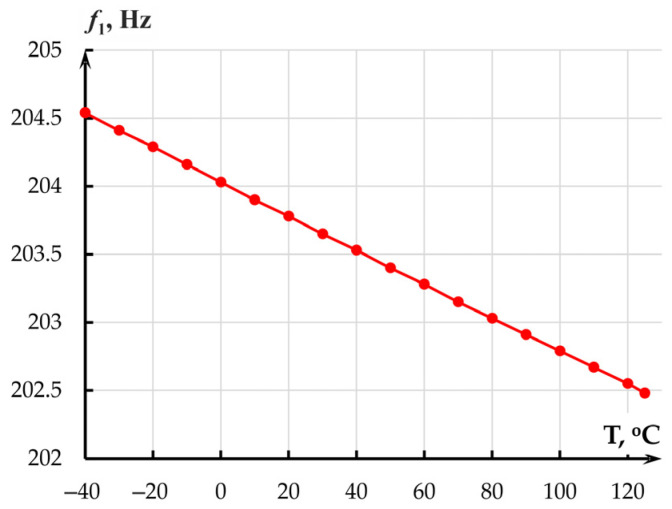
Dependence of eigenfrequency *f*_1_ on temperature.

**Figure 8 sensors-25-06388-f008:**
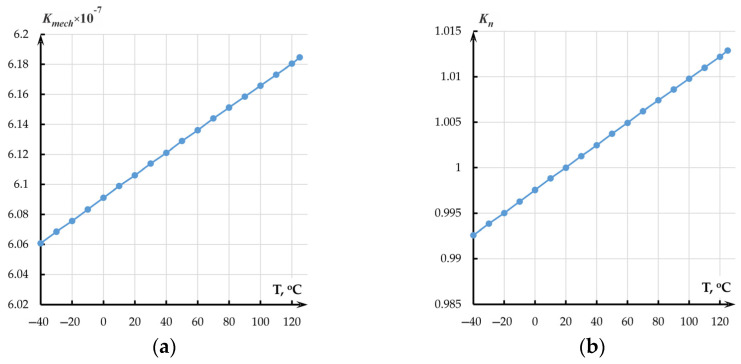
Dependence of the mechanical transmission coefficient of the accelerometer on temperature: (**a**) Absolute values; (**b**) Normalized transmission coefficient.

**Figure 9 sensors-25-06388-f009:**
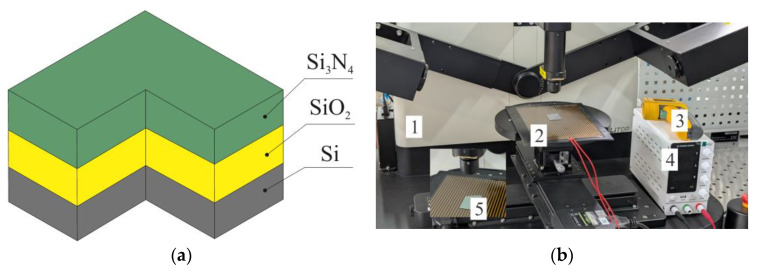
Experimental stand for determining the values of temperature coefficients of the refraction index: (**a**) Graphical representation of the films; (**b**) Experimental setup.

**Figure 10 sensors-25-06388-f010:**
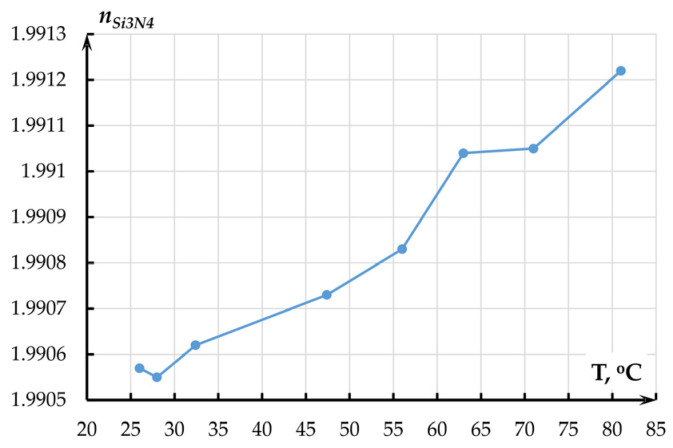
Temperature dependence of refraction index of silicon nitride (Si_3_N_4_) film on the SiO_2_/Si substrate.

**Figure 11 sensors-25-06388-f011:**
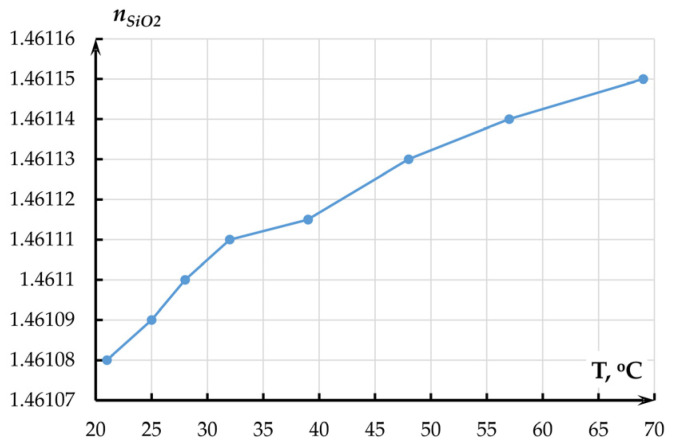
Temperature dependence of the refraction index of the SiO_2_ film on the Si substrate.

**Figure 12 sensors-25-06388-f012:**
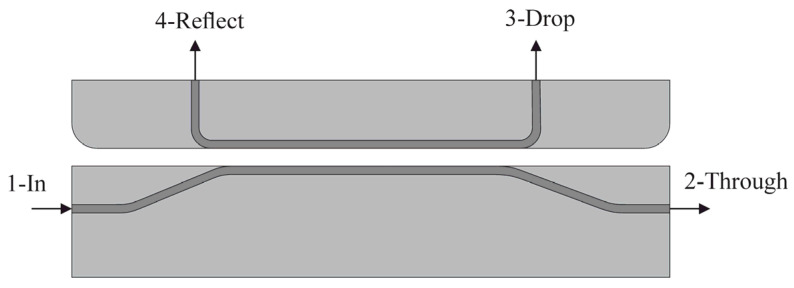
Model of the OMT based on the directional coupler.

**Figure 13 sensors-25-06388-f013:**
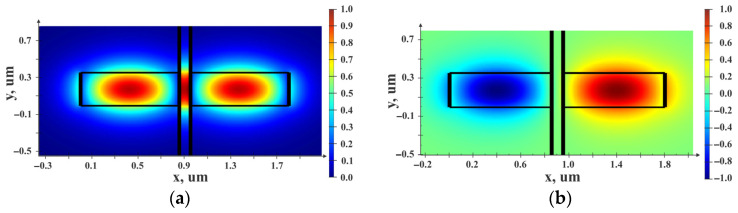
Distribution of the optical field in the directional coupler with 350 × 850 nm waveguides: (**a**) Even mode; (**b**) Odd mode.

**Figure 14 sensors-25-06388-f014:**
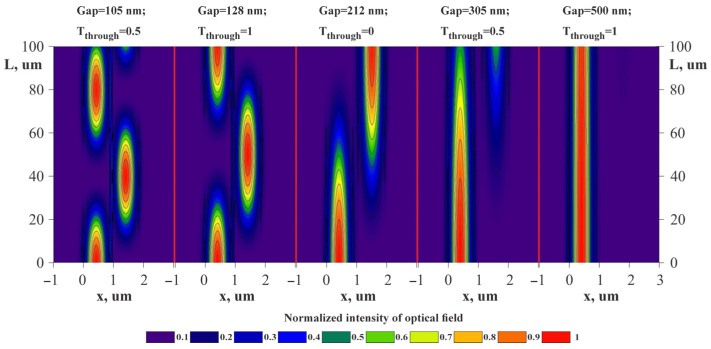
Power maps at different gaps obtained by the FDE method.

**Figure 15 sensors-25-06388-f015:**
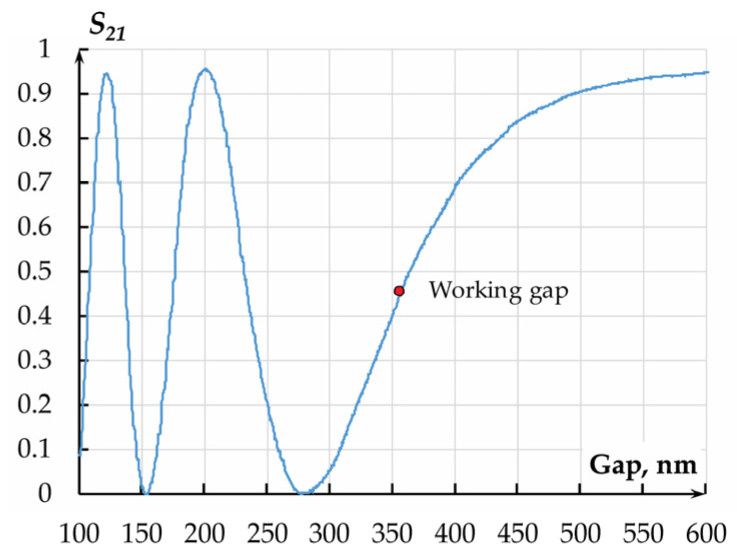
Dependence of the OMT’s *S*_21_ on the gap at a wavelength of 1550 nm obtained by the FDTD method.

**Figure 16 sensors-25-06388-f016:**
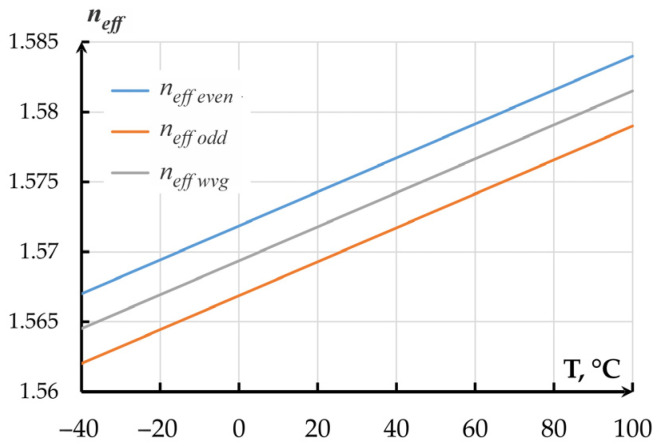
Dependence of the effective indexes of the directional coupler and 350 × 850 nm waveguide on temperature.

**Figure 17 sensors-25-06388-f017:**
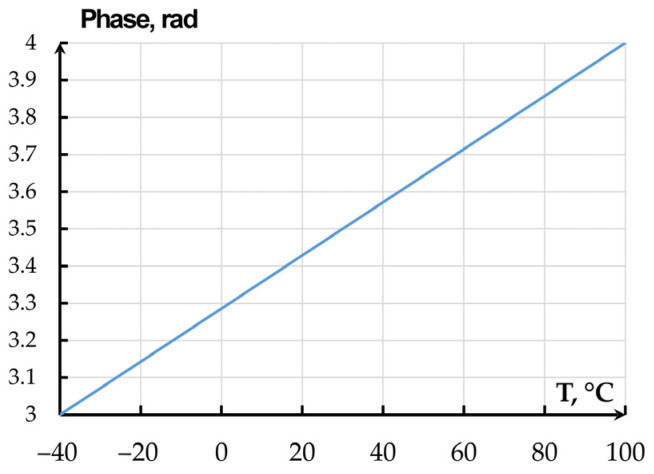
Dependence of the 350 × 850 nm waveguide phase shift on temperature.

**Figure 18 sensors-25-06388-f018:**
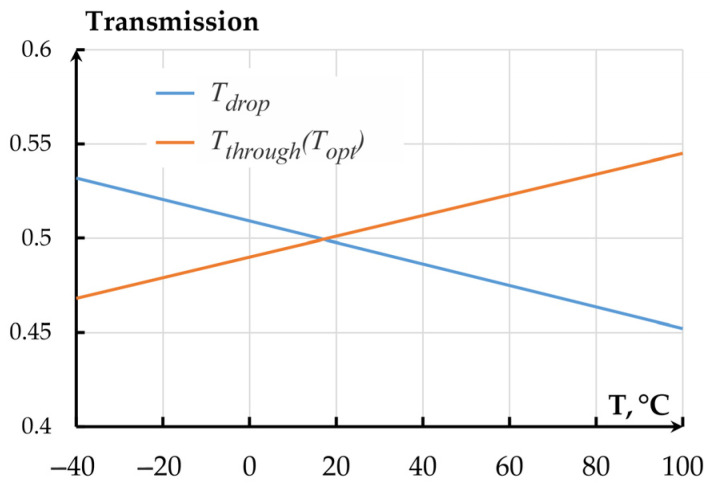
Dependence of the OMT’s optical transmission coefficient on temperature.

**Figure 19 sensors-25-06388-f019:**
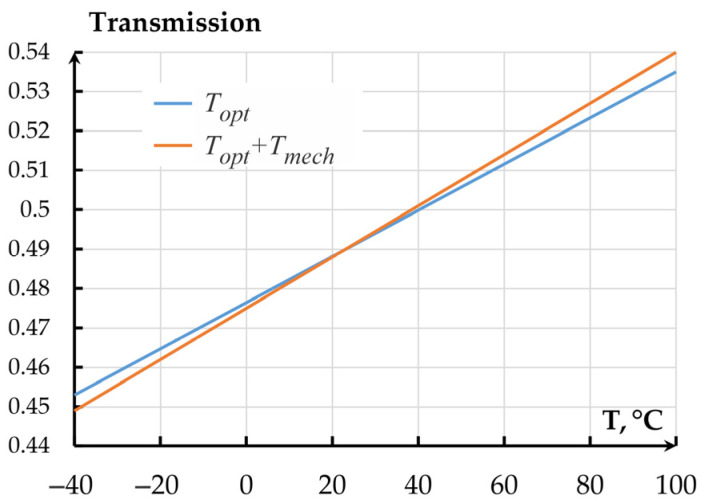
Dependence of the MOEMA’s total transmission coefficient on temperature.

**Figure 20 sensors-25-06388-f020:**
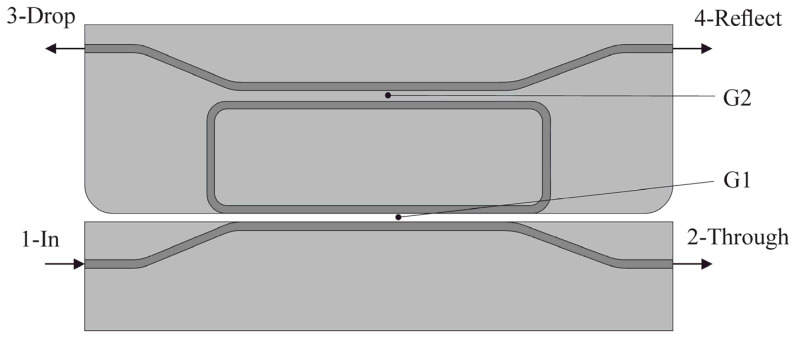
Model of the accelerometer’s resonator-based OMT.

**Figure 21 sensors-25-06388-f021:**
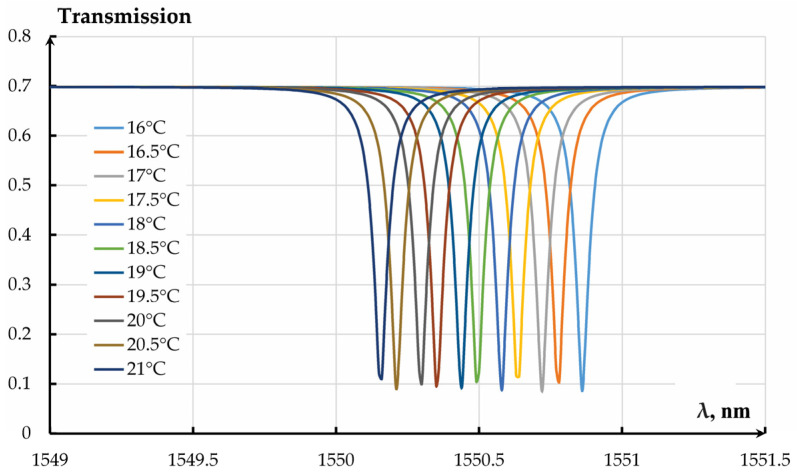
Amplitude frequency response of OMT resonator at different wavelength.

**Figure 22 sensors-25-06388-f022:**
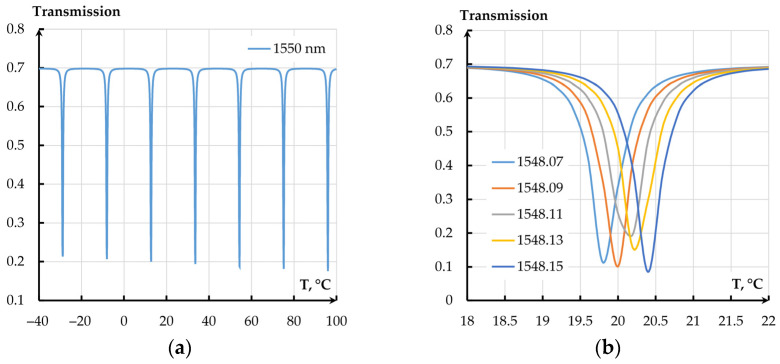
Dependencies of the optical transmission coefficient of the resonator-based OMT on temperature: (**a**) Temperature range from minus 40 to plus 100 °C; (**b**) Temperature range from 18 to 23 °C.

## Data Availability

Data are contained within the article.

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
