# Peer review of "Thermal Sensitivity of a Microoptoelectromechanical Evanescent-Coupling-Based Accelerometer"

_sensors, 2025, doi:10.3390/s25206388_

Round 1

Reviewer 1 Report

Comments and Suggestions for Authors

The paper presents “Thermal sensitivity of a microoptoelectromechanical evanescent-coupling-based accelerometer”. However, this paper might be acceptable for publication if the following questions should be answered properly.

  • The paper presents the influence of temperature on the dynamic characteristics of an evanescent coupling-based MOEM accelerometer. But the influence of temperature on the mechanical transmission coefficient maybe analyzed too idealized. The residual stress induced in package processing may be large than that of the mechanical parts of sensor devices. So, the package of the MOEM accelerometers should be added and analyzed.
  • The paper presents some testing results of the influence of temperature on the optical transmission coefficient. The more detail of the theoretical and FEM simulated analysis should be added.
  • The more detail of the testing setting of the influence of temperature on the optical transmission coefficient should be added.
  • The whole devices testing results of the influence of temperature should be added.

Author Response

Comment 1: 

The paper presents the influence of temperature on the dynamic characteristics of an evanescent coupling-based MOEM accelerometer. But the influence of temperature on the mechanical transmission coefficient maybe analyzed too idealized. The residual stress induced in package processing may be large than that of the mechanical parts of sensor devices. So, the package of the MOEM accelerometers should be added and analyzed.

Response 1:

You are correct. The residual stresses arising during packaging may considerably affect the mechanical transmission coefficient. The severity of this effect is determined by a chosen packaging technology. In the review (line 108, reference [26]), we have noted the presence of such effect. In current article, we mainly focused on optical measuring transducers of an accelerometer; the mechanical part was considered only in terms of the die. The proposed question will be scrutinized in our further research, and the results will depend on the selection of the package and die encapsulation technology.

Comment 2:

The paper presents some testing results of the influence of temperature on the optical transmission coefficient. The more detail of the theoretical and FEM simulated analysis should be added.

Response 2:

We agree. The necessary information on the modelling process was added and equation 5 was amended (lines 324-327, 350-355, 376).

Comment 3:

The more detail of the testing setting of the influence of temperature on the optical transmission coefficient should be added.

Response 3:

We agree. Additional data describing the measurement conditions for the refraction index temperature dependence were added (lines 295-298).

Comment 4:

The whole devices testing results of the influence of temperature should be added.

Response 4:

The article presents the results of a numerical study of a device based on experimentally obtained temperature dependencies of refraction coefficients for the materials in use (SiO2 and Si3N4). For the whole device testing proposed by you and adequate comparison of experimental and theoretical data, the refraction index temperature coefficient should be measured for separate waveguides of a specific device design. Direct measurements on an ellipsometer (like it was made for films) are impossible due to the ellipsometer’s laser spot dimensions—above 5 µm—while the waveguides are 1 µm wide. These measurements can be made indirectly, for instance by the resonance method as described in references [1–3] below the answer. The results of indirect measurements on the MOEM structure will take into account all the influence factors (mechanical, optical, residual stresses, etc.). Separating them into components and comparing with the modeling results is extremely difficult from methodological perspective or even impossible. For example, work [4] concluded that there are no numerical methods for calculation and prognostication of internal stresses induced by a technological process of thin film growth. The experiments on waveguides fabricated on a separate wafer will be unrepresentative in terms of true physics in a MOEM structure due to different mechanical parts. Therefore, both internal stresses and stresses induced by thermal expansion of the materials will also be different.

The studies proposed by you require extensive research, novel techniques and comprehensive description, and are beyond the scope of the present work. We plan to work on these issues in further studies.

  1. Bogaerts, P. De Heyn, T. Van Vaerenbergh, K. De Vos, S. Kumar Selvaraja, T. Claes, P. Dumon, P. Bienstman, D. Van Thourhout, and R. Baets, «Silicon microring resonators», Laser Photonics Review, Vol. 6, No. 1, pp. 47-73 (2012). https://doi.org/10.1002/lpor.201100017
  2. Vivien, and L. Pavesi, Handbook of Silicon Photonics, 1st Edition, edited by L. Vivien, L. Pavesi, (CRC Press, 2020), p. 868. https://doi.org/10.1201/b14668
  3. R. McKinnon, D.-X. Xu, C. Storey, E. Post, A. Densmore, A. Delage, P. Waldron, J. H. Schmid, and S. Janz, «Extracting coupling and loss coefficients from a ring resonator», Optics Express, Vol. 17, No. 21, pp. 18971-18982 (2009). https://doi.org/10.1364/OE.17.018971
  4. Shugurov, A.R., Panin, A.V. Mechanisms of Stress Generation in Thin Films and Coatings. Tech. Phys. 65, 1881–1904 (2020). https://doi.org/10.1134/S1063784220120257

Reviewer 2 Report

Comments and Suggestions for Authors

The authors investigate how non-ideal characteristics of an electric motor influence vibration dynamics in a three-degree-of-freedom (3-DOF) system, particularly focusing on nonlinear phenomena associated with the Sommerfeld effect. This area remains of scientific and engineering relevance. The manuscript has the potential to contribute to the field. However, several critical and structural issues must be addressed before it could be considered for publication.

Major Concerns:

- The manuscript contains multiple grammatical errors, awkward phrasing, and typos that impede clarity. A thorough language revision, preferably by a native speaker or professional service, is essential.

- The rationale for employing a 3-DOF system is not persuasive. The phenomena observed—primarily the Sommerfeld effect, could likely be captured adequately using a single-DOF model. The authors need to justify explicitly why the added complexity of a 3-DOF system yields novel insights not accessible in simpler configurations.

- Beyond introducing the 3-DOF model, the manuscript lacks a clear statement detailing its unique contribution relative to existing literature. The authors should articulate what is new, whether it’s the modeling framework, specific parameter regimes, or interpretation of the Sommerfeld effect in multi-DOF contexts.

- The manuscript does not specify which numerical method was used to solve Eq. (11), nor does it describe the solution workflow. Please provide details on: The integration algorithm (e.g., Runge-Kutta, variable-step solver, etc.), Convergence criteria, time stepping, stability checks, Initial conditions and parameter sweep procedures, A visual flowchart or pseudo-algorithm would greatly enhance transparency and reproducibility.

-  No validation of the numerical results is presented. To build confidence in the findings: Compare results with analytical solutions (if available), benchmark against existing literature, or validate with experimental or high-fidelity simulation data. At minimum, include a sensitivity or convergence study.

- The parameters for unbalance mass (m) and eccentricity (r) appear separately, yet their combined effect (product m·r) governs the magnitude of unbalance torque. Consider simplifying by using m·r as a single variable and base variations on recognized standards, such as ISO 1940-1.

- The current parameter set appears idealized. The authors should broaden the study by introducing more realistic case studies, reflecting typical stiffness, damping, unbalance, and scale found in practical systems, to assess the engineering relevance of the Sommerfeld effect under real-world conditions.

Minor Concerns:

-  Expand citations of previous studies on multi-DOF models and experimental observations of the Sommerfeld effect to position the current work within the field.

- Ensure all figures have clear axes labels, units, and concise captions. Some plots are difficult to interpret due to dense formatting.

- Ensure all equations are referenced and cross-referenced consistently (e.g., parenthetical numbering).

-  Enhance the discussion to contextualize the findings—what do these results imply for system design, control, or prediction of multi-DOF mechanical oscillators?

In summary, and based on the issues outlinedabove, particularly regarding novelty, methodological transparency, validation, and presentation, I recommend major revisions. Addressing these points would substantially improve the manuscript’s clarity, rigor, and scientific contribution.

Author Response

Response:

We have familiarized ourselves with your review. We are afraid, the comments do not correspond to this article.

Reviewer 3 Report

Comments and Suggestions for Authors

This manuscript presents a comprehensive investigation of the thermal sensitivity of a microoptoelectromechanical accelerometer based on evanescent coupling. The study addresses a key challenge for optical MEMS devices: performance degradation across a wide operational temperature range (-40°C to +125°C). The authors correctly highlight a research gap in the literature concerning accelerometers utilizing directional couplers and ring resonators, establishing the work's novelty.  The manuscript is well-structured and well-written. There are several typos, which do not hinder understanding, but careful proofreading is necessary. The reference list is extensive and includes recent publications. The conclusions logically follow from the presented results and fully address the research questions set out in the study's objectives.

I just have few suggestions

  1. The abstract mentions a range from -40°C to +125°C, while mechanical simulations (Figures 4-8) use -65°C to +125°C. Please clarify or align these ranges.
  2. For the ring resonator, the thermal sensitivity (0.33 °C⁻¹) is ~3000 times larger than the mechanical contribution. This extreme sensitivity renders it unusable without precise temperature control, confining it to a thermostatic environment. I recommend to add brief discussion.

Author Response

Comment 1:

The abstract mentions a range from -40°C to +125°C, while mechanical simulations (Figures 4-8) use -65°C to +125°C. Please clarify or align these ranges.

Response 1:

We agree. The Figures were amended 5, 6, 7, 8a, 8b.

Comment 2:

For the ring resonator, the thermal sensitivity (0.33 °C⁻¹) is ~3000 times larger than the mechanical contribution. This extreme sensitivity renders it unusable without precise temperature control, confining it to a thermostatic environment. I recommend to add brief discussion.

Response 2:

We have elaborated the section end (Measuring transducer based on a resonator) (lines 423-425) and the beginning of Discussion section (lines 427-430).

Round 2

Reviewer 2 Report

Comments and Suggestions for Authors

Acceptable in the current version...